# Age Related Differences in Monocyte Subsets and Cytokine Pattern during Acute COVID-19—A Prospective Observational Longitudinal Study

**DOI:** 10.3390/cells10123373

**Published:** 2021-11-30

**Authors:** Anita Pirabe, Stefan Heber, Waltraud C. Schrottmaier, Anna Schmuckenschlager, Sonja Treiber, David Pereyra, Jonas Santol, Erich Pawelka, Marianna Traugott, Christian Schörgenhofer, Tamara Seitz, Mario Karolyi, Bernd Jilma, Ulrike Resch, Alexander Zoufaly, Alice Assinger

**Affiliations:** 1Institute of Vascular Biology and Thrombosis Research, Center of Physiology and Pharmacology, Medical University of Vienna, Schwarzspanierstrasse 17, 1090 Vienna, Austria; anita.pirabe@meduniwien.ac.at (A.P.); waltraud.schrottmaier@meduniwien.ac.at (W.C.S.); anna.schmuckenschlager@meduniwien.ac.at (A.S.); n01504003@students.meduniwien.ac.at (S.T.); david.pereyra@meduniwien.ac.at (D.P.); jonas.santol@meduniwien.ac.at (J.S.); ulrike.resch@meduniwien.ac.at (U.R.); 2Institute of Physiology, Center of Physiology and Pharmacology, Medical University of Vienna, 1090 Vienna, Austria; stefan.heber@meduniwien.ac.at; 3Department of Surgery, Division of Visceral Surgery, Medical University of Vienna, General Hospital Vienna, 1090 Vienna, Austria; 4Department of Medicine IV, Clinic Favoriten, 1010 Vienna, Austria; Erich.Pawelka@gesundheitsverbund.at (E.P.); marianna.traugott@gesundheitsverbund.at (M.T.); tamara.seitz@gesundheitsverbund.at (T.S.); mario.karolyi@gesundheitsverbund.at (M.K.); alexander.zoufaly@gesundheitsverbund.at (A.Z.); 5Department of Clinical Pharmacology, Medical University of Vienna, General Hospital Vienna, 1090 Vienna, Austria; christian.schoergenhofer@meduniwien.ac.at (C.S.); bernd.jilma@meduniwien.ac.at (B.J.); 6Faculty of Medicine, Sigmund Freud University, 1020 Vienna, Austria

**Keywords:** inflammaging, immunoparalysis, COVID-19, aging, monocytes, innate immune response

## Abstract

The COVID-19 pandemic drastically highlighted the vulnerability of the elderly population towards viral and other infectious threats, illustrating that aging is accompanied by dysregulated immune responses currently summarized in terms like inflammaging and immunoparalysis. To gain a better understanding on the underlying mechanisms of the age-associated risk of adverse outcome in individuals experiencing a SARS-CoV-2 infection, we analyzed the impact of age on circulating monocyte phenotypes, activation markers and inflammatory cytokines including interleukin 6 (IL-6), IL-8 and tumor necrosis factor (TNF) in the context of COVID-19 disease progression and outcome in 110 patients. Our data indicate no age-associated differences in peripheral monocyte counts or subset composition. However, age and outcome are associated with differences in monocyte activation status. Moreover, a distinct cytokine pattern of IL-6, IL-8 and TNF in elderly survivors versus non-survivors, which consolidates over the time of hospitalization, suggests that older patients with adverse outcomes experience an inappropriate immune response, reminiscent of an inflammaging driven immunoparalysis. Our study underscores the value, necessity and importance of longitudinal monitoring in elderly COVID-19 patients, as dynamic changes after symptom onset can be observed, which allow for a differentiated insight into confounding factors that impact the complex pathogenesis following an infection with SARS-CoV-2.

## 1. Introduction

Inflammaging, an age-associated chronic inflammatory state, has been linked to increased incidents of infections and decreased responses to vaccines in the aging population. In the current COVID-19 pandemic, the importance of age-related processes became especially apparent as older adults account for a disproportionate number of severe cases and deaths, with patients between 75 and 84 having a 220-fold and patients above 85 having a 570-fold higher mortality risk compared to 18- to 29-year-old individuals [1].

Inflammaging arises due to degeneration of immune receptors which lose specificity and affinity, thus failing to discriminate self- from non-self-ligands, leading to uncontrolled, stochastic activation of the innate immune system. It can therefore be regarded as a long-term result of chronic inflammation with concurrent activation of the innate immune system, which eventually becomes self-destructive during aging [2].

Monocytes are a heterogeneous cell population of the mononuclear phagocyte system, which is a major component of innate immunity. These cells are considered the main producers of cytokines following inflammatory stimuli and danger signals, and thus have long been speculated as the major contributors to inflammaging [3]. While monocyte functions are known to become dysregulated during aging, the phenotypic and functional differences between monocyte subsets during homeostasis and various disease conditions are only partly understood.

In humans, monocytes are generally described as three subsets based on their relative expression of CD14 and CD16 [4]. However, additional phenotypes based on gene and protein expression patterns have been identified [5,6]. The relative proportions of these subtypes are associated with a variety of diseases, with increasing severity generally linked to increased prevalence of the CD16+ intermediate and non-classical subsets [7,8].

While little phenotypic differences were observed in unstimulated monocytes obtained from older individuals, stimulated monocytes showed impaired transcriptional and biological responses compared to those of younger individuals [9], supporting the notion that aging influences the cytokine profile of monocytes and their agonist-provoked-responses [10].

SARS-CoV-2 can infect monocytes through angiotensin converting enzyme 2 (ACE2)-dependent and independent pathways. In any case, infected circulating monocytes exert inappropriate activities that contribute to excessive cytokine production, widespread tissue damage and death [11].

In this study we aimed at elucidating age-related changes of circulating peripheral monocytes during acute COVID-19. We characterized monocyte subsets in 110 patients and monitored them during their hospital stay. In parallel, we examined cytokine patterns in these patients.

## 2. Materials and Methods

### 2.1. Participants and Inclusion Criteria

This study was carried out as part of the Austrian Coronavirus Adaptive Clinical Trial (ACOVACT; ClinicalTrials.gov NCT04351724). The study was conducted under approval of the local ethics committee (EK1315/2020) and carried out in accordance with the Declaration of Helsinki. Between 17 April 2020 and 28 October 2020, patients admitted to the Clinic Favoriten participated and gave informed consent. A total of 110 patients between 18 and 92 years of age were enrolled. The inclusion criteria were confirmed SARS-CoV-2 infection by real-time PCR of naso- or oropharyngeal swab and ≥18 years of age. Exclusion criteria were life expectancy under one month (e.g., due to severe comorbidities), pregnancy or breast feeding, anemia (hemoglobin < 11 g/dL), stage-4 kidney disease and severe liver dysfunction.

### 2.2. Data and Sample Collection

A medical history was taken at the day of hospital admission. COVID-19 severity was classified according to the World Health Organization (WHO) scores into mild, moderate, severe and critical. Clinical outcomes including uncomplicated, intensive care unit (ICU) and death were documented up to 21 days after symptom onset. Clinical parameters were assessed at hospital admission, or latest 72 h after admission. Blood samples for plasma preparation and flow cytometric analyses were collected at the day of hospital admission (day 0) or at the latest 9 days after admission followed by repeated blood draws every 2–3 days over the first week and then every 7 days. Only samples collected no later than 21 days after symptom onset were included in the analyses (Figure 1A, Appendix A).

### 2.3. Sample Preparation

Blood was drawn into vacutainer tubes containing citrate or CTAD. Plasma of CTAD-anticoagulated blood was generated by centrifugation at 1000× *g* for 10 min at 4 °C, followed by a second centrifugation of the platelet-free supernatant at 10,000× *g* for 10 min. Plasma samples were aliquoted and stored at −80 °C until analysis without further freeze and thaw cycles.

### 2.4. Flow Cytometry

Citrate-anticoagulated whole blood was prepared at the latest 3 h after blood draw for flow cytometry analysis. The blood of 97 patients were stained with fluorescently-labeled antibodies (anti-CD66b-Pacific Blue, anti-CD16-PE-Cy7, anti-CD11b-activated-FITC; (all BioLegend), antiCD14-APC (BD Biosciences)) for 20 min and diluted with 1-step Fix/Lyse solution (eBioscience) (Appendix A) to deplete for erythrocytes. Samples were measured on a Cytoflex S cytometer within 6 h and analysed using CytExpert 2.4 software (both Beckman Coulter). Leukocytes were identified based on the FSC and SSC properties, followed by the exclusion of doublets and multiplets. Monocytes were classified as CD66b-negative and CD14-positive singlet leukocytes. Monocyte subpopulations were characterized by the expression of CD14 and CD16. For both markers, 3000 monocytes were recorded. Activated monocytes were defined by the expression of activated CD11b on their surface (Appendix A). Monocyte subsets were quantified as a percentage of the total population (%) and activated CD11b as mean fluorescence intensity (MFI).

### 2.5. Cytokine Analysis

Plasma cytokine and chemokine levels (IL-6, IL-8, TNF and MCP-1) of 108 patients were quantified using a LEGENDplex^TM^ bead-based immunoassay kit (BioLegend) according to the manufacturer’s instructions. Briefly, plasma was incubated with capture beads for 2 h while mixing. After washing, samples were incubated with biotinylated detection antibodies for 1 h forming capture bead-analyte-detection antibody complexes. Subsequently, streptavidin-phycoerythrin (SA-PE) was added for 30 min providing a fluorescent signal. Samples were measured on a Cytoflex S cytometer within 3 h and analyzed using LegendPlex v8.0 software (BioLegend). Absolute concentrations of IL-6, IL-8, TNF and MCP-1 (pg/mL) were determined in relation to a standard concentration curve.

### 2.6. Statistical Analysis

Statistical analyses were performed with IBM SPSS Statistics 28, diagrams were generated with GraphPad Prism 8, Adobe Illustrator CS6 160.0, Microsoft Excel 2016 and SankeyMATIC.

Nominal variables were compared using the Chi-square test, normal-distributed metric data were compared using the (2-sided) Student’s *t*-Test and non-normal-distributed metric data were compared using the (2-sided) Mann Whitney U-Test. The Kolmogorov-Smirnov Test was applied to assess for normality. Those *p*-values ≤ 0.05 were considered as statistically significant.

Separate linear mixed models for each readout were applied to explore if the cytokine/-, chemokine levels and monocyte subset ratios develop differently over the disease course in younger and elderly COVID-19 patients and whether this development in turn differs between the outcome groups, i.e., uncomplicated course, ICU requirement and death.

Depending on the data distribution, values were either left untransformed, log10- or logit-transformed. The predictors ‘age’ in years and ‘days after symptom onset’ were used as continuous covariates. For each of these predictors, an additional quadratic term was included to allow for non-linear effects. ‘Outcome’ was included as a factor with three levels, namely uncomplicated, ICU or death. Each subject was included as level of a random factor allowing a random intercept for each patient.

At first, a mixed linear model was calculated with all main effects and all possible interactions including ‘age’, ‘days after symptom onset’ and ‘outcome’ (i.e., 3 main effects, 3 two-way interactions and the 3-way interaction) plus two quadratic terms of ‘age’ and ‘days after symptom onset’. Next, non-significant interactions were omitted from the model beginning with the 3-way interaction, followed by two-way interactions beginning with the one having the highest *p*-value. This procedure was performed until only significant interactions or only main effects were left. Least-square means (with 95% confidence intervals) were computed to allow visualization of the final models including the uncertainty of estimates. Before plotting, estimated values were back-transformed which allows plotting on the original scale.

3D plots were generated to illustrate the central tendencies of variables (either means or geometric means of back-transformed logit values) for 21 days post-symptom onset with ages ranging from 20 to 85 years. In addition, the model was sliced at ages of interest and measures of central tendencies were plotted with confidence intervals for the clinical outcomes uncomplicated, ICU and death. The *p*-values ≤ 0.05 were considered significant. Only two-sided tests were used.

## 3. Results

Analysis of our patient cohort revealed that outcome was not determined by WHO severity classification at admission, neither in the whole cohort, nor in the age-based or sex-based separated groups (Figure 1B, Appendix A). However, patients above the age of 75 less frequently experienced symptoms like dyspnoea (*p* = 0.002) and fever (*p* = 0.044) compared to younger COVID-19 patients (Figure 1C, left), but no striking differences in presentation of symptoms were observed in patients above 75 years that survived compared to non-survivors (Appendix A, left). In addition, the prevalence of SARS-CoV-2 related symptoms did not differ between female and male patients (Appendix A, left). As expected, older patients had a higher, but not significantly increased, incidence of comorbidities, in particular cardiovascular diseases, malignancies and chronic obstructive pulmonary disease (COPD) (Figure 1C, right). COPD (*p* = 0.013) was more frequent in non-survivors compared to survivors in the same age group (Appendix A, right). Comorbidities were less frequently observed in women that required intensive care but more often observed in female patients with an unfavorable outcome when compared to men. Especially malignancies and hypertension were more frequent among deceased women (Appendix A). When we analyzed disease severity according to patient age, we found that patients of all age groups experienced mild to critical symptoms upon admission (Figure 1D). However, when we analyzed outcome in the patient cohort, we found that only patients above 75 years died (Figure 1E). However, sex specific analysis revealed that the age distribution of female and male patients did not vary significantly in the different outcome groups. However, the number of male patients requiring ICU treatment was higher compared to female patients (Appendix A, right). Based on our cohort demography, we stratified our subsequent analysis into two groups, below and above 75 years of age.

Next, we analyzed changes of circulating neutrophil and monocyte counts over 21 days after symptom onset according to the patient’s age groups. However, no clear trend of neutrophil counts in relation to disease onset could be observed (Figure 2A, above). Similarly, we found no age-specific differences in monocyte counts, suggesting that not the absolute numbers, but rather their functional responses may differ with age (Figure 2A, below). Therefore, we analyzed age-specific differences in the expression of monocyte surface markers discriminating classical (CD16-negative) from non-classical (CD16-positive) subsets over the course of disease (Figure 2B, left). We found a clear decrease in classical monocytes (*p* = 0.001) (Figure 2B, middle, Appendix A) and an increase in non-classical monocytes (*p* < 0.0001) (Figure 2B, right, Appendix A) over time independent of age (classical monocytes: age-dependent time effect *p* = 0.50, time-independent age effect *p* = 0.70, non-classical monocytes: age-dependent time effect *p* = 0.20, time-independent age effect *p* = 0.034).

To determine outcome specific differences in monocyte subsets we analyzed monocyte patterns over time in regard to patient outcome, grouped into uncomplicated, ICU and death (Figure 3A). We found no statistically significant evidence that the pronounced decrease of classical monocytes (main effect of time *p* < 0.0001) was different between outcome-groups over time (outcome-dependent time effect *p* = 0.24). Nonetheless, groups differed significantly independent of time (*p* = 0.034) (Figure 3A). When we cut the 3D graphs plotted in Figure 3A at the patient’s age of 75 years, 80 years and 85 years (Figure 3B), it became apparent that classical monocytes were slightly more frequent in ICU patients (Figure 3B, middle) (contrast independent of time or age, ICU vs. uncomplicated *p* = 0.011) but not in non-survivors (contrast independent of time or age, death vs. uncomplicated *p* = 0.83). In turn, non-classical monocytes increased over the course of disease, but barely showed any differences between outcome groups (all interactions with time or age *p* > 0.09, main effect of outcome *p* = 0.50, Figure 3C,D).

We then determined levels of monocyte activation makers including MCP-1, a proinflammatory chemokine induced downstream of the angiotensin II signaling cascade, and surface expression of the activated integrin adhesion receptor CD11b in COVID-19 patients (Figure 4A, left). Projection of MCP-1 levels over disease progression and patient age revealed lowest levels in middle-aged, followed by young and highest in elderly patients (Figure 4A, middle and Appendix A) (quadratic term, corresponding to a non-linear effect of age *p* = 0.032). Additionally, MCP-1 levels slightly decreased in all three age groups over the disease course (*p* = 0.002) (Figure 4C). Levels of activated CD11b did not display this concave-like age distribution, but gradually decreased with age and during course of disease (Figure 4A, right). However, even after transforming the data, there is no evidence for different CD11b activation between patients below and above 75 years of age (Appendix A).

When we analyzed for outcome specific differences (Figure 4B,D), we found that MCP-1 decreased over time independent of outcome (outcome-specific time effect *p* = 0.13). Highest levels of circulating MCP-1 were found in older individuals with an unfavorable outcome (Figure 4B,C), indicating that MCP-1 levels not only increase with age (quadratic term, corresponding to a non-linear effect of age *p* = 0.034), but further increase in fatal COVID-19 cases (age- and time independent outcome effect corresponding to a general difference between groups *p* = 0.002, Contrasts: ICU vs. uncomplicated *p* = 0.004, death vs. uncomplicated *p* = 0.03). When we analyzed activated CD11b levels according to outcome, we found no apparent age-related changes in patients with uncomplicated disease or patients requiring ICU treatment. However, ICU patients and patients who did not survive showed reduced CD11b activation compared to COVID-19 patients with an uncomplicated course of disease (Figure 4D,E).

We then analyzed three major inflammatory cytokines reportedly associated and possibly causally involved in COVID-19 pathology irrespective of outcome, namely IL-6, IL-8 and TNF. While plasma levels of IL-6 and IL-8 appeared increased with age (IL-6 *p* = 0.017, IL-8 *p* = 0.003) (Figure 5A and Appendix A), surprisingly, TNF was not age-associated (*p* = 0.72) (Figure 5A). When comparing TNF between two age groups, we found no age-associated differences in TNF plasma levels (Appendix A).

When we analyzed IL-6 levels over time according to age and outcome, we found a significant decrease over the course of disease in uncomplicated and ICU patients (*p* < 0.001), but not in non-survivors (Figure 5B). In general, patients with uncomplicated outcome exhibit the lowest levels of IL-6, followed by ICU treated patients and highest levels of IL-6 being measured in non-survivors. At 80 years of age, where the dataset allows comparison of all three outcomes, IL-6 levels did not differ by outcome at symptom onset (*p* = 0.51) (Figure 6A, middle). However, estimated plasma IL-6 levels at 21 days post symptom onset suggest significant group differences (*p* < 0.001), with patients displaying an uncomplicated course of disease having lower IL-6 levels compared to the other two groups (uncomplicated vs. ICU *p* < 0.001 and uncomplicated vs. death *p* < 0.001).

Further, patients with uncomplicated course of disease show a rapid decline of IL-6 over time, while non-survivors show persistent high levels of IL-6 (Figure 6A).

Regarding IL-8 levels, we found a significant decrease following symptom onset (*p* < 0.001) and age (*p* = 0.003), (Figure 5C). However, time- and age-specific effects did not affect each other (*p* = 0.23), suggesting that the relative decrease in IL-8 is not influenced by age. When we plotted the estimated geometric IL-8 means according to outcome, time and age, levels appeared to rapidly drop in patients with an uncomplicated course of disease, while in patients requiring ICU treatment IL-8 only dropped in younger, but not in elderly patients (Figure 5C). Highest IL-8 levels were observed in non-survivors, where levels remained high over time (Figure 5C and Figure 6B). Statistically, this is reflected by an interaction between days after symptom onset and outcome (*p* < 0.001), indicating that IL-8 levels develop in dependence of outcome, illustrated by a significant interaction *p*-value (*p* = 0.04) between the days after symptom onset and age, implying that the IL-8 levels over time were additionally dependent on age.

Finally, we analyzed circulating levels of TNF in our COVID-19 patient cohort. While TNF levels were generally independent of age or days after symptom onset (Figure 5A, *p* > 0.44 each), patient stratification according to outcome revealed that the impact of age is more pronounced in patients that required ICU treatment compared to patients without complications (Figure 5D, outcome-specific age effect *p* = 0.036). When we compared patients of the same age group according to outcome, TNF levels were lowest in non-survivors (Figure 6C).

## 4. Discussion

Inflammaging plays a central role in dysregulated immune responses. As monocytes are key players in this process, we aimed to gain a better understanding on the underlying mechanism of the age-associated risk of adverse outcome in individuals experiencing SARS-CoV-2 infection. We analyzed the impact of age on circulating monocyte phenotypes and inflammatory cytokines in plasma in the context of COVID-19 disease progression and outcome. Data obtained from our patient cohort do not support the presumption that differences in peripheral monocyte counts or phenotypic subset composition are associated with age. In contrast, we found that monocyte functionalities, as determined by representative activation markers, are particularly affected by age and additionally associated with COVID-19 disease outcome. As a consequence, a dysregulated immune response, which aggravates over time, results in a distinct inflammatory cytokine pattern of IL-6, IL-8 and TNF in elderly survivors versus non-survivors.

While older patients experienced less severe symptoms compared to younger patients, only patients above the age of 75 died in our study cohort. As symptoms, including cough, dyspnoea, fever and several more assessed in this study, are often a result of an activated immune system, this observation suggest that older patients potentially have a dysregulated immune response, which recapitulates features of immunoparalysis. As comorbidities are naturally perceived as being increased with age, we found that hypertension was associated with adverse outcome only when we analyzed patients of the same age group. This is in line with previous reports on inappropriate and weak immune response appearing more frequently in patients with comorbidities, thereby facilitating viral spreading and disease severity [12].

The current SARS-CoV-2 pandemic increased the awareness of sex-specific differences in COVID-19 associated immunity and clinical outcomes. In our cohort the number of male patients with uncomplicated disease and those requiring intensive care was much higher compared to female COVID-19 patients, whereas the number of deceased patients did not vary between sexes. This is in line with other studies, which reported that men become more severely ill compared to women [13]. In addition, it was found that the mortality rate among men were elevated, with the highest fatality rate observed among men with comorbidities [14,15]. Moreover, comorbidities including hypertension, diabetes, chronic respiratory diseases and cardiovascular diseases, which are associated with a severe COVID-19 disease progression, were found to be more frequent among men [13]. This was not reflected in our cohort, since comorbidities including hypertension, obesity or malignancies, which are associated with a severe disease. Inflammatory markers, ACE2 concentration as well as markers of liver and kidney function are higher in male COVID-19 patients [15,16], were observed more frequently in deceased female patients. The reasons for sex-associated differences in COVID-19 are still unclear. Sex-based biological differences, which affect cell-mediated immunity and antibody production as well as gender-based behavioral differences such as smoking and handwashing are discussed [17]. We also analyzed sex-specific differences in our patient cohort but found no statistically significant association between sex and outcome.

Contrary to our initial hypothesis, we found no differences in neutrophil or monocyte counts between young and older patients, neither upon admission nor during their hospital stay, indicating that not leukocyte quantities, but rather their functions are impaired.

Analysis of monocyte subsets revealed a clear decrease in classical monocytes along with increased non-classical monocytes over the course of disease, but this was independent of age. While there was no difference in patients with uncomplicated disease and non-survivors, ICU-treated patients showed slightly higher levels of classical monocytes. This is in line with previous reports [18], which also observed an increase in a distinct CD16 positive monocyte subpopulation, while another study found lower monocytic CD16 levels predictive for disease severity, although patient outcome was not analyzed [19]. The observed increase in CD16 positive monocytes is not specific to SARS-CoV-2 infection as patients suffering from other viral or bacterial infections reportedly show a similar increase in CD16 expressing monocytes [20]. This supports the hypothesis that SARS-CoV-2 drives monocytes and macrophages to induce host immunoparalysis, which, potentially translates into less severe symptoms, reduced MHC class 2 expression and TNF production as cells become refractory to chronic stimulation for the benefit of COVID-19 progression [21]. However, no age specific effects on peripheral monocyte subtypes could be observed. This is surprising, as aging has been associated with an increased quantity of non-classical monocytes in conjunction with higher levels of plasma TNF and IL-8, conditions which have previously been discussed as contributors to inflammaging [22].

To get a better understanding on monocyte chemotaxis and activation, we analyzed circulating monocyte chemoattractant protein-1 (MCP-1) levels as well as CD11b activation on the monocyte surface. MCP-1 is produced by several cell types, including macrophages and fibroblasts upon direct stimulation with pattern recognition receptors (PRR) and by cytokines such as IL-6 and TNF [23]. MCP-1 acts as a chemoattractant for monocytes and is essential for routine immunological surveillance of tissues, as well as in response to inflammation [24]. Lack of MCP-1 results in more severe disease progression in response to viral infections [25].

Circulating MCP-1 levels in COVID-19 patients increased with age revealing the highest MCP-1 levels in older individuals with an unfavorable outcome, indicating that not only age, but also unfavorable outcome is cause or consequence of increased MCP-1. As COVID-19 progressed, MCP-1 levels slightly decreased but this effect was outcome independent. Activation of the integrin receptor CD11b on the other hand decreased with age and patients requiring ICU treatment as well as patients with adverse outcome had even lower levels of activated CD11b, further supporting the notion of dysregulated monocyte polarization and monocyte paralysis in severe disease courses of COVID-19. A limited functionality of monocytes was also reported by another study, in which reduced expression of CD86, CD40 and a high percentage of PD-L1 on classical monocytes was associated with reduced monocyte activation and T cell stimulation. As a consequence, antibody-mediated immunity is reduced, which in turn enhances the risk of viral reactivation [26].

An overshooting production of inflammatory mediators, termed “cytokine storm”, represents a major threat in complicated COVID-19 and a plethora of cytokines have been demonstrated to be crucially involved [27]. Although monocytes are important in this process, recent evidence suggests that peripheral monocytes do not express substantial amounts of pro-inflammatory cytokines in patients with COVID-19 [28]. In line with recently published research, we found an age dependent increase of IL-6 and IL-8 in our COVID-19 cohort [29].

Further, IL-6 levels were lowest in patients with uncomplicated disease, followed by ICU treated patients. Non-survivors showed the highest IL-6 levels. Importantly, when we analyzed patients of the same age, IL-6 levels did not differ by outcome at symptom onset. However, patients with an uncomplicated disease course show a rapid decline of IL-6 over time, while non-survivors show persistently high levels of IL-6. A previous study found that the MHC-2 expression on monocytes and lymphocytes were strongly inhibited by plasma from COVID-19 patients with high IL-6 levels, supporting the notion that an overproduction of IL-6 contributes to disease progression [30].

IL-8 levels rapidly dropped independent of age as COVID-19 progressed. IL-8 levels appeared to drop more rapidly in patients with an uncomplicated disease course, while in patients requiring ICU treatment IL-8 only dropped in younger but not in elderly patients. Highest IL-8 levels were observed in non-survivors where levels remained high over time. Of note, one difference between IL-6 and IL-8 was that at admission, patients with uncomplicated disease showed similar or even higher IL-8 plasma levels compared to ICU patients. This might explain why IL-6, but not IL-8, was predictive for outcome in some cohorts but failed to show significant differences in other cohorts. Based on our data one could speculate that IL-6 alterations occur earlier in the natural disease course than the ones of IL-8.

TNF levels were generally independent of age and days after symptom onset. When we compared patients from the same age group, non-survivors had lower TNF levels compared to patients with uncomplicated course or ICU-treated patients. This is surprising, as previous studies showed that both IL-6 and TNF levels represent independent and significant predictors of disease severity and death even after adjustment for age [31]. In addition, TNF were found to promote T cell apoptosis, which is possibly why TNF levels inversely correlate with T cell counts in COVID-19 patients requiring intensive care [32].

Important to note, our study population differs from previously described COVID-19 patient cohorts. First, this Austrian cohort shows a wider age distribution compared to other cohorts. Secondly, since Austria never reached limits in hospitalization and ICU capacities, patients were never triaged, resulting in a wide distribution of disease severities and even patients without symptoms were registered in hospitals. In Austria a total of 81,787 COVID-19 cases were reported, with 45,188 patients being hospitalized and 655 deaths between 17 April and 28 October 2020. Patients were often hospitalized earlier as compared to patients in other countries, allowing for monitoring over longer periods. Since patients were admitted at different disease stages, we stratified all patients for days after onset of symptoms to allow for a better comparison. However, this is prone to a reporting bias. Another limitation of the study is the relatively small sample size that does not allow for further subgroup analyses and the partially high number of missing clinical parameter values. Further a control group of healthy age matched participants or patients with other infections would have been beneficial for the interpretation of our results. Especially controls with similar comorbidities, would be important to get a better understanding of the immunopathogenesis in these patients to unravel COVID-19 specific changes and effects of other inflammatory diseases. Moreover, we cannot rule out that some patients had multiple infections and since age and outcome are often associated, we cannot always dissect clearly, if differences are due to age or outcome. Despite these limitations, our study sheds light on the prevailing question: what makes elderly patients more prone to suffer from adverse outcome after COVID-19 infection.

Understanding the precise immune responses—also in respect to age, sex and co-morbidities—represents a pre-requisite for successful therapeutic interventions.

Immunomodulatory agents such as corticosteroids successfully decrease mortality in COVID-19 patients [33]. However, more targeted approaches including inhibition of IL-6 and TNF signaling have been suggested. While many trials are still ongoing, a recent meta-analysis revealed, that therapeutic IL-6 (receptor) antagonists are effective in reducing mortality in COVID-19 patients, while the risk of side effects is higher [34], highlighting the potential but also the need for better therapies.

Taken together, our data indicated no age associated differences in peripheral monocyte counts or subset composition, but age and outcome are reflected on monocyte activation status. Moreover, a distinct cytokine pattern of IL-6, IL-8 and TNF in elderly survivors versus non-survivors which aggravates over time suggests that elderly patients experience an inappropriate immune response, reminiscent of a phenomenon termed immunoparalysis. Many previous studies correlated circulating cytokines, often assessed as “snap-shot” sampling at single or a limited number of days during disease progression, according to the WHO severity score. However, our data show that a severe COVID-19 disease state at admission does not necessarily imply adverse outcome. Hence, our study underscores the value, necessity and importance of monitoring patients over a period of time, as dynamic changes after symptom onset can be observed which allow for a differentiated insight into confounding factors which impact the complex outcome following an infection with SARS-CoV-2.

## Figures and Tables

**Figure 1 cells-10-03373-f001:**
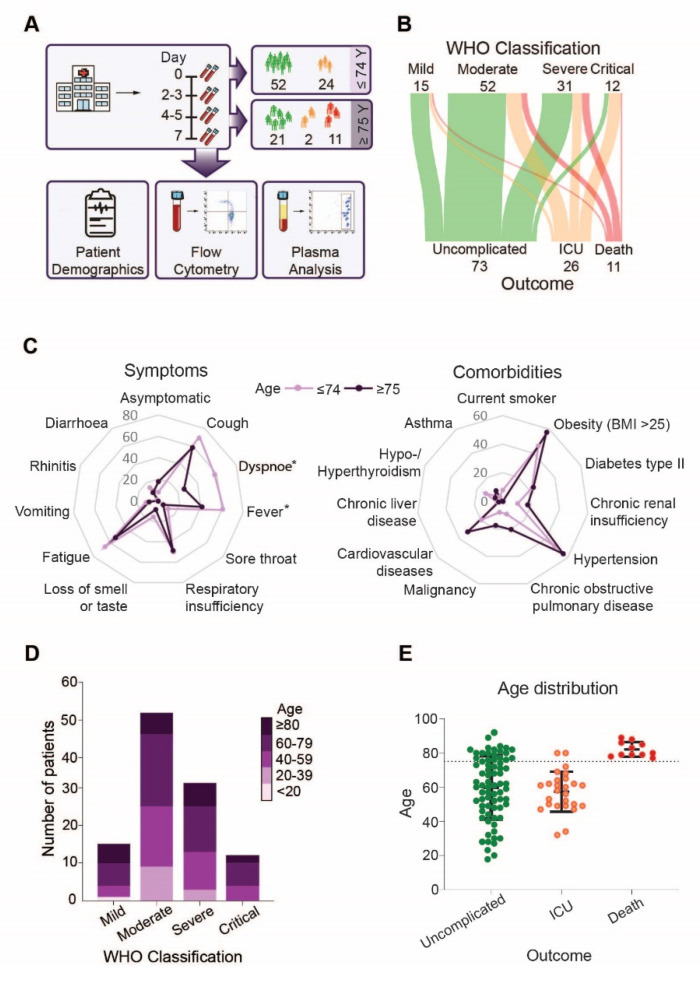
Disease severity, symptoms and outcome in younger and elderly COVID-19 patients: (**A**) Overview of the study design, (**B**) Sankey diagram visualizing WHO disease severity classification at hospital admittance and clinical outcome of COVID-19 patients, (**C**) Lines in the spider chart show the percentage of patients below and above 75 years of age having symptoms (**left**) or comorbidities (**right**), (**D**) Disease severity according to age, (**E**) Age distribution in the different outcome groups. Asterisk indicate significant differences between patients below and above the age of 75 years.

**Figure 2 cells-10-03373-f002:**
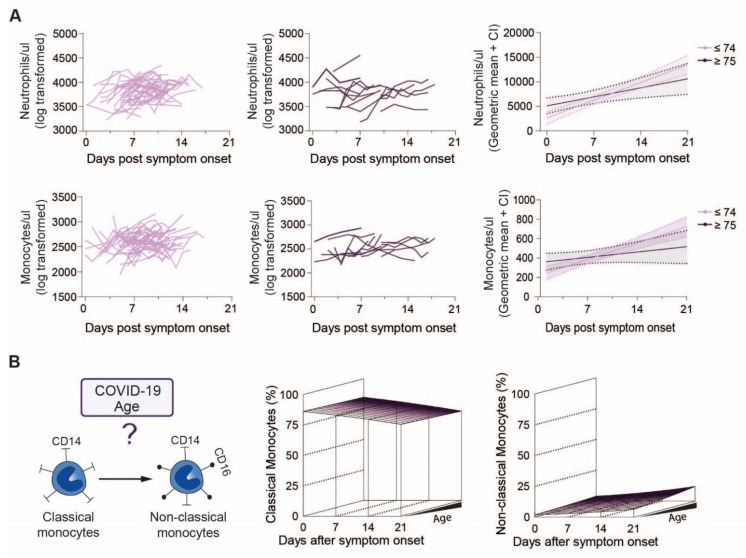
Neutrophil and monocyte counts of younger and elderly COVID-19 patients over the disease course: (**A**) Raw values of neutrophils and monocytes of individual COVID-19 patients below (**left**) or above (**middle**) 75 years of age and modelled values (**right**) of the two age groups over time, (**B**) Study overview of monocyte subpopulations in COVID-19 patients regarding age (**left**) and modelling of classical (**middle**) and non-classical (**right**) monocytes in patients over the disease course.

**Figure 3 cells-10-03373-f003:**
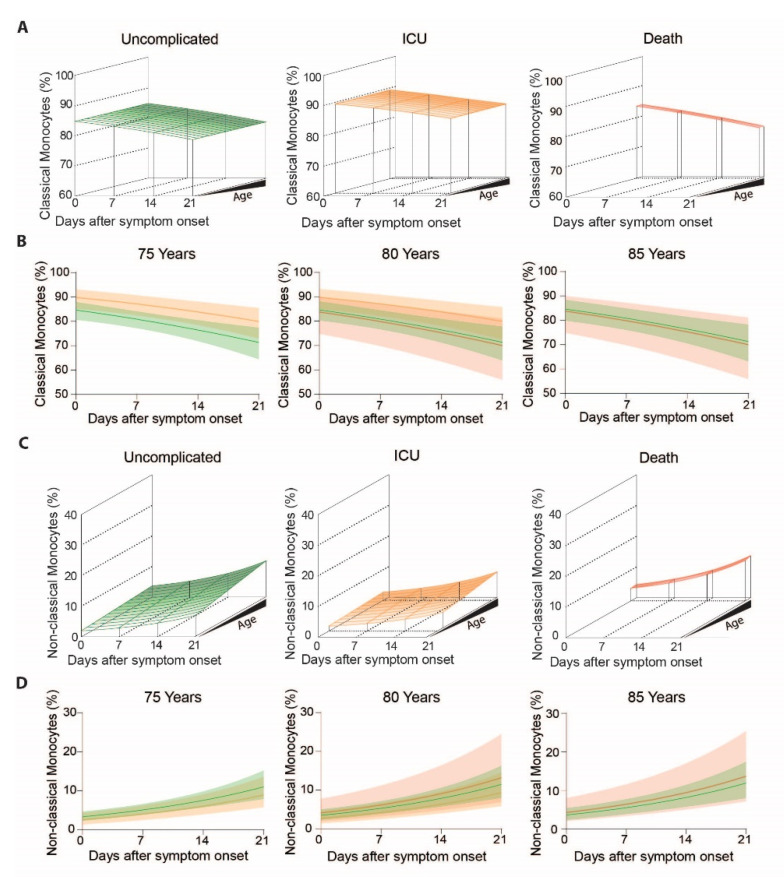
Classical and non-classical monocyte subset frequencies of COVID-19 patients with uncomplicated disease, requiring ICU or adverse outcome over the disease course: (**A**) Modelled classical monocyte frequencies in COVID-19 patients with uncomplicated disease (**left**), requiring ICU (**middle**) and adverse outcome (**right**) according to age and time, (**B**) Means with confidence intervals of modelled classical monocytes according to the clinical outcome at 75 (**left**), 80 (**middle**) and 85 (**right**) years of age, (**C**) Modelled non-classical monocyte frequencies in COVID-19 patients with uncomplicated disease (**left**), requiring ICU (**middle**) and adverse outcome (**right**) according to age and time, (**D**) Means with confidence intervals of modelled non-classical monocytes according to the clinical outcome at 75 (**left**), 80 (**middle**) and 85 (**right**) years of age.

**Figure 4 cells-10-03373-f004:**
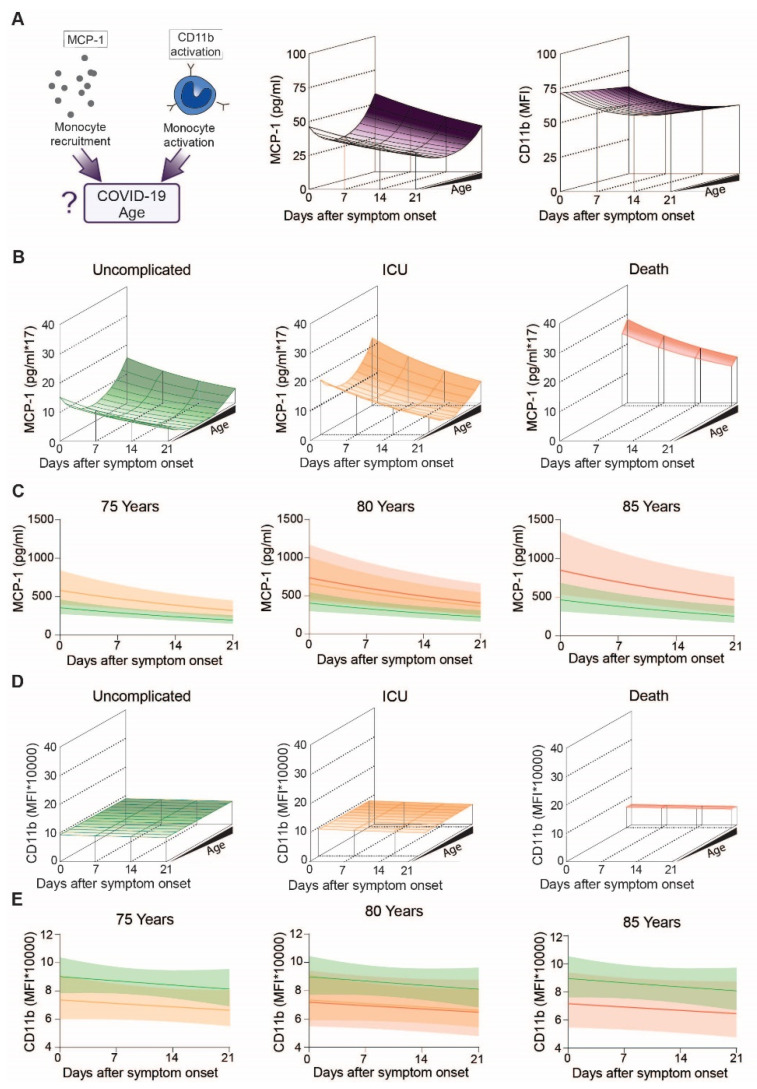
Specific markers for monocyte stimulation in COVID-19 patients with uncomplicated disease, requiring ICU or adverse outcome: (**A**) Study overview of MCP-1 and activated CD11b regarding age (**left**), Modeling of MCP-1 (**middle**) and CD11b activation (**right**) levels of COVID-19 patients over the disease course, (**B**) Modeling of MCP-1 of patients with uncomplicated disease (**left**), requiring ICU (**middle**) or adverse outcome (**right**), (**C**) Means with confidence intervals of modeled MCP-1 according to the clinical outcome at the age of 75 (**left**), 80 (**middle**) and 85 (**right**) years, (**D**) Modeling of CD11b activation patients with uncomplicated disease (**left**), requiring ICU (**middle**) or adverse outcome (**right**), (**E**) Means and confidence intervals of modelled CD11b activation according to outcomes at the age of 75 (**left**), 80 (**middle**) or 85 (**right**) years.

**Figure 5 cells-10-03373-f005:**
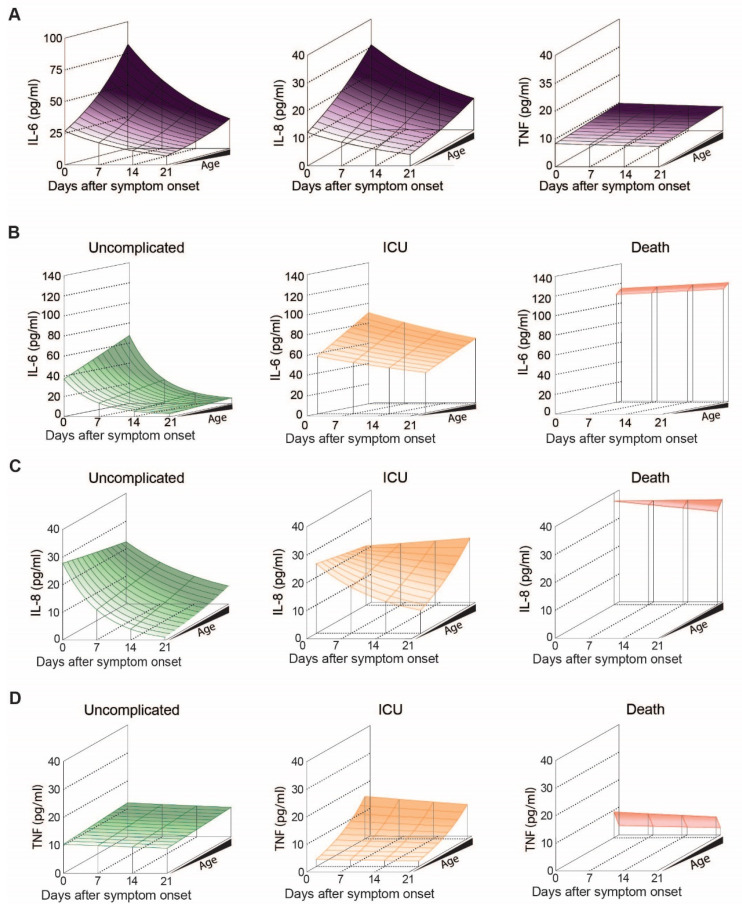
IL-6, IL-8 and TNF levels in COVID-19 patients with uncomplicated disease, requiring ICU or adverse outcome: (**A**) Modelled plasma levels of IL-6 (**left**), IL-8 (**middle**) and TNF (**right**) of patients aged between 20 and 85 years over the disease course post symptom onset, (**B**) Modelled plasma levels of IL-6, (**C**) IL-8 and (**D**) TNF in patients with uncomplicated disease, requiring ICU or adverse outcome.

**Figure 6 cells-10-03373-f006:**
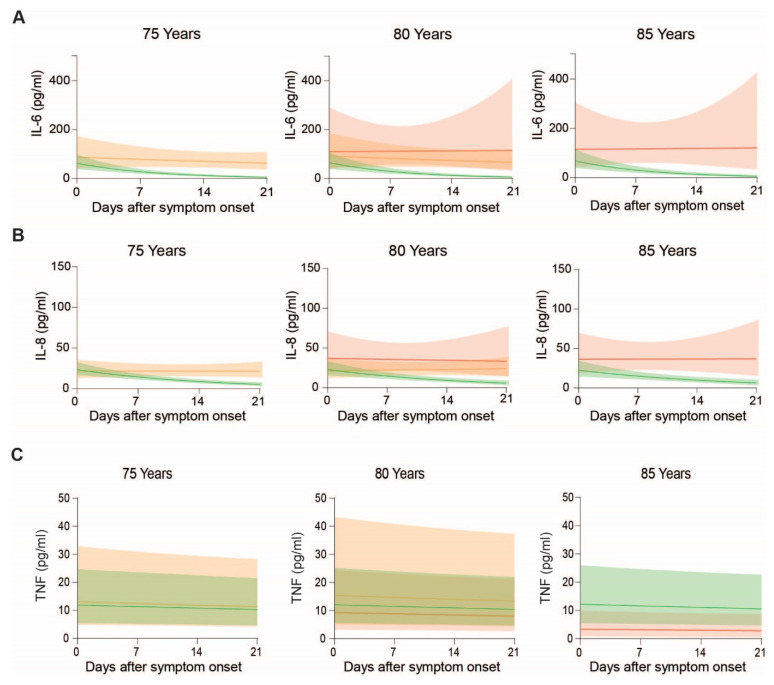
Plasma IL-6, IL-8 and TNF levels in COVID-19 patients over the disease course at the age of 75, 80 and 85 years: Means and confidence intervals of modeled (**A**) IL-6, (**B**) IL-8 and (**C**) TNF of patients with uncomplicated disease, requiring ICU and adverse outcome.

## Data Availability

The data that support the findings of this study are available from the corresponding author upon reasonable request.

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
