# Peer review of "Age Related Differences in Monocyte Subsets and Cytokine Pattern during Acute COVID-19—A Prospective Observational Longitudinal Study"

_cells, 2021, doi:10.3390/cells10123373_

Round 1
Reviewer 1 Report
Please, find comments in the attached PDF file.

Author Response
Reviewer 1:
- Please provide samples of participation consent of patients of your study in the supplementary file.
Reply: We have added a participation consent to the supplementary file.
- Please, add a record about the incidence of COVID-19 in Austria. This will clear for the reader the severity of the disease at the time of your study.
Reply: We thank the reviewer for this comment and have added a paragraph in the discussion. In Austria a total of 81 787 COVID-19 cases were reported, with 45 188 patients being hospitalized and 655 died between 17th of April and 28th of October 2020. Although Austria was hit hard during the first wave of the pandemic, we never reached the limit of our hospital capacity and no patients were triaged. Of note, Austria is the country with the second highest number of hospital beds per capita, therefore patients were admitted early and all patients monitored in hospital even when the numbers peaked.
- Please categorize the age to young, medium and old.
Reply: We thank the reviewer for this suggestion. Although at first sight categorization would simplify the interpretability of the study results and is therefore often employed in clinical research, there is a plethora of literature arguing against categorization of continuous predictors (Altman and Royston 2006, Royston, Altman et al. 2006, Bennette and Vickers 2012)
To name a few of the problems, categorization of continuous metric variables can negatively affect the overall predictive ability of a statistical model. Furthermore it can lead to a loss of statistical power and introduce bias. Another issue is the choice of cut-off values and the number of categories generated. These choices severely affect p-values regarding the comparison of categories. In the worst case, so-called “optimal” cut-off values are data-driven, which opens the door for p-hacking and can lead to serious bias. Furthermore, categorization implicitly assumes homogeneity within risk groups, which is obviously inappropriate in most cases. Therefore, continuous variables should generally be included in statistical models in their original (continuous) form. Based on that we decided not to categorize the predictor “age”.
Altman, D. G. and P. Royston (2006). "The cost of dichotomising continuous variables." Bmj 332(7549): 1080.
Bennette, C. and A. Vickers (2012). "Against quantiles: categorization of continuous variables in epidemiologic research, and its discontents." BMC Medical Research Methodology 12(1): 21.
Royston, P., D. G. Altman and W. Sauerbrei (2006). "Dichotomizing continuous predictors in multiple regression: a bad idea." Stat Med 25(1): 127-141.
- Please divide the title of the three (sample preparation, flow cytometry and cytokine analysis). Write some details about flow cytometry conditions that were used in your experiment.
Reply: We thank the reviewer for this suggestion and have separated the three sections and revised the experimental details on the flow cytometric experiments.
- As you mentioned in the supplementary table, you collected the data of male and female, please describe and discuss differences between male and female.
Reply: This is an excellent suggestion of the reviewer. We have analyzed the effect of gender on outcome in a Chi-square test. However this was not significant (p = 0.301). As outlined in the limitation section our sample size is limited and we do not want to over-interpret our results. However, we added a supplementary figure (Supplementary Figure 2) to describe sex differences with regard to outcome and age. And as suggested by reviewer 3 also comorbidities. Moreover, we added a paragraph discussing sex-related differences in COVID-19 to the discussion.
- Where is the data of degree of COVID-19 severity and your study blood parameters. Where is the flow cytometry histogram results?
Reply: The data on the degree of COVID-19 severity is found in Figure 1B, presented as a Sankey blot as well as in all details together with the blood parameters in Supplemental Table 1. We can also move the table into the main manuscript if preferred. Regarding the histograms of the flow cytometry data, we have added Supplemental Figure 5 which shows the gating strategy as also suggested by reviewer 2.
- Where is the data of other ages.
Reply: As only patients above 75 years of age died we decided to visualize only these patients in more detail. Therefore we did “cut” our graphs every 5 years to allow for comparison between ages (75, 80 and 85).
- The age scale in 3D plots in all figures is uncertified. I suggest to categorize your study age to young, medium and old and compare between them. Also make 3D plots of young, medium and old ages
Reply: We thank the reviewer for this remark. As mentioned in #3 we prefer not to categorize our data due to statistical reasons. We agree that the exact age is hard to see in a 3D graph. However, we intended only to visualize if a parameter increases or decreases with age. For detailed visualization we did cut the graphs at age 75, 80 and 85, as only patients of these age group showed adverse outcome.
- Discussion: In this section the authors didn´t discuss differences between male and female.
Reply: This in an important issue raised by the reviewer. As mentioned already in 5. we do not see a statistical significant association between outcome and gender. This could also be due to the relative small sample size. Although we added a descriptive Supplemental Figure on sex-specific differences, we would like to interpret the data with care as we have only a small cohort. However, we now added more literature on the current knowledge on sex related differences in COVID-19 patients to the revised discussion.
- Contrary to our expectations: Please change this sentence. It is a study.
Reply: We thank the reviewer for pointing out this mistake, which is now corrected to “Contrary to our initial hypothesis….”
- Are you ensure these increased levels of IL-6 and IL-8 is related to the age COVID-19 not to other mixed viral infection.
Reply: We thank the reviewer for this important comment. Indeed we cannot rule out that patients also have multiple infections (viral and bacterial) and discuss this now in the limitation section of the discussion.
- Please discuss these results with published studies with updated references.
Reply: We have undated the discussion and now include more references on the pathophysiology of COVID-19, sex-related aspects and anti-inflammatory treatment options.
Reviewer 2 Report
In this paper authors analyzed monocyte profile in patients with SARS-CoV2 infection. The paper is well written, data are clearly presented and conclusions are supported by the results.
Minor concerns:
- Why did you used a whole blood for flow cytometry analysis, and you didn’t lyse the erythrocytes prior to labeling with antibodies?
- It would be good to show representative FSC-SSC scatter in supplementary files and to note the population that you marked as leukocytes
- Authors should correct a mislabeling in Figure 2A. Upper right diagram is mislabeled; instead of monocytes it should be neutrophils.
Author Response
In this paper authors analyzed monocyte profile in patients with SARS-CoV2 infection. The paper is well written, data are clearly presented and conclusions are supported by the results.
Minor concerns:
- Why did you used a whole blood for flow cytometry analysis, and you didn’t lyse the erythrocytes prior to labeling with antibodies?
Reply: We apologize for this lack of clarity. We did lyse erythrocytes after antibody labelling using a fix and lyse kit (1-step Fix/Lyse solution (eBioscience)). We describe this now more clearly in the revised Material and Methods Section.
- It would be good to show representative FSC-SSC scatter in supplementary files and to note the population that you marked as leukocytes
Reply: We thank the reviewer for his/her suggestion to add a representative histogram that shows our gating strategy, which is now found in Suppl. Figure 5.
- Authors should correct a mislabeling in Figure 2A. Upper right diagram is mislabeled; instead of monocytes it should be neutrophils.
Reply: We thank the reviewer for pointing out this mistake, which is now corrected.
Reviewer 3 Report
I am not including these comments to the authors. There are 3 major issues which cannot be corrected with modifications. If helpful I can report this to the authors as well.
- The inclusion and exclusion criteria are not well thought out. Diabetes and cardiovascular disease are known to be significant co-morbidities that were not considered as exclusion criteria. These and other variables contributing to disease development make it very difficult to make concrete conclusions on the dependent variables.
- This needs to be a case control study and no controls were included (age controls and healthy controls). This is a major study design flaw.
- Their findings in cell types and cytokine levels may be signatures of disease outcomes, but are not prognostic nor suggestive of treatment success. There is also little discussion in immunopathogenesis associated with their findings. Without this carefully developed discussion the findings are just results with no added understanding in the concept of inflammaging.
Author Response
I am not including these comments to the authors. There are 3 major issues which cannot be corrected with modifications. If helpful I can report this to the authors as well.
- The inclusion and exclusion criteria are not well thought out. Diabetes and cardiovascular disease are known to be significant co-morbidities that were not considered as exclusion criteria. These and other variables contributing to disease development make it very difficult to make concrete conclusions on the dependent variables.
Reply: This is a very important issue. We are fully aware that comorbidities have a significant contribution. Therefore we carefully evaluated them and added this information to the revised manuscript and further analyse them in Supplemental Figure 2 of the revised manuscript. However, if we would have excluded these patients our sample size would be very low. Therefore we decided to obtain samples from as many patients as possible and analyse comorbidities as confounding factors. We clearly discuss this in the revised manuscript and added the fact that we lack COVID-19 negative controls with similar comorbidities to the limitation section.
- This needs to be a case control study and no controls were included (age controls and healthy controls). This is a major study design flaw.
Reply: We completely agree with the reviewer that this would have been important for the interpretation of our results. Unfortunately in the heat of the pandemic we were unable to collect and monitor age matched healthy controls. We have now added this as a limitation in the discussion of the revised manuscript.
- Their findings in cell types and cytokine levels may be signatures of disease outcomes, but are not prognostic nor suggestive of treatment success. There is also little discussion in immunopathogenesis associated with their findings. Without this carefully developed discussion the findings are just results with no added understanding in the concept of inflammaging.
Reply: We thank the reviewer for this important comment. We would like to state that our study is only hypothesis generating and was not designed to develop a predictive biomarker or suggest treatment options. We also do not claim this in the manuscript. We also thank the reviewer for his/her critical comment on the discussion, we have revised the discussion and now add more information on the immunopathogenesis associated with our findings.
Round 2
Reviewer 1 Report
The authors consider all comments and I am satisfied for this study.